# A Retrospective Chart Review of Factors Impacting Psychotropic Prescribing Patterns and Polypharmacy Rates in Youth with Autism Spectrum Disorder during the COVID-19 Pandemic

**DOI:** 10.3390/jcm11164855

**Published:** 2022-08-18

**Authors:** Evan Taniguchi, Kerry Conant, Kylie Keller, Soo-Jeong Kim

**Affiliations:** 1Psychiatry and Behavioral Sciences, University of Washington, Seattle, WA 98195, USA; 2Psychiatry and Behavioral Sciences, Seattle Children’s Hospital, Seattle, WA 98105, USA; 3Seattle Children’s Autism Center, Seattle Children’s Hospital, Seattle, WA 98105, USA

**Keywords:** autism, psychotropic, polypharmacy, demographic, psychiatry, COVID-19

## Abstract

High but variable rates of psychotropic polypharmacy (PP) in youth with autism spectrum disorder (ASD) have been reported in previous studies. The effect of the COVID-19 pandemic on prescribing patterns has not been well described. This study aims to examine the factors associated with psychotropic prescribing patterns, including rates of PP and multiclass polypharmacy (MPP) in youth with ASD during the COVID-19 pandemic. We examined the prescription records and clinical characteristics of youth aged between 3–21 years with a clinical diagnosis of ASD who were followed at an urban tertiary autism center psychiatry clinic between 1 January 2019, and 31 December 2020. For study purposes, we treated 2019 as the pre-pandemic year and 2020 as the pandemic year and compared the clinical characteristics of the “total clinic cohort *(n =* 898)” across two years. We examined the clinical characteristics of patients seen in both years (“paired-sample,” *n* = 473) and those seen only in 219 (“not-paired sample,” *n* = 378) to identify factors associated with the likelihood of patients’ return to clinic in 2020. As the total clinic cohort was a naturalistic sample containing duplicate patients, we created a separate data set by randomly assigning duplicate patients to one of the years (“random unique sample,” *n* = 898) and examined the clinical characteristics across two years. We defined PP and MPP broadly as the use of ≥2 unique medications (PP) and ≥2 unique medication classes (MPP) within a calendar year in this study. In the total clinic cohort, increased rates of PP (71.6% to 75.6%), MPP (61.9% to 67.8%, *p* = 0.027), and antidepressant prescriptions (56.9% to 62.9%, *p* = 0.028) were noted, although only the latter two were nominally significant. The paired-sample had a higher proportion of teens (31.0% vs. 39.7%, *p* < 0.001 and persons who self-identified as non-Hispanic (77.8% vs. 85.4%, *p* = 0.016)), higher rates of anxiety (78.9% vs. 48.7%, *p* < 0.001), ADHD (71.0% vs. 44.4%, *p* < 0.001), depression (23.9% vs. 13.0%, *p* < 0.001) and disruptive behavior (63.3% vs. 33.3%, *p* < 0.001) diagnoses, higher rates of antidepressants (63.4% vs. 48.7%, *p* < 0.001), ADHD medications (72.5% vs. 59.8%, *p* < 0.001), and antipsychotics (36.8% vs. 26.2%, *p* < 0.001) prescribed, and higher rates of PP (81.6% vs. 59.0%, *p* < 0.001) and MPP (71.0% vs. 50.5%, *p* < 0.001) than the not-paired sample. In the random unique sample, the patient group assigned to 2020 had higher rates of anxiety (75.0% vs. 60.2%, *p* < 0.001), ADHD (69.9% vs. 54.6%, *p* < 0.001), and disruptive behavior (57.9% vs. 45.4%, *p* < 0.001) diagnoses but the PP and MPP rates did not differ across years. Overall, we found high rates of PP and MPP, likely due to the broader definition of PP and MPP used in this study than those in other studies as well as the study site being a tertiary clinic. While our study suggests a possible impact of the COVID-19 pandemic on comorbidity rates and prescribing patterns, a replication study is needed to confirm how pandemic-related factors impact prescribing patterns and polypharmacy rates in youth with ASD.

## 1. Introduction

The core symptoms of autism spectrum disorder (ASD) include social communication deficits, restricted interests, and repetitive behaviors. Many individuals with ASD are affected by other emotional and behavioral problems, commonly referred to as “problem behaviors” such as aggression, irritability, and self-injury, as well as other medical and psychiatric comorbidities [1]. Past studies have suggested that 10–70% have at least one comorbid psychiatric disorder with the most common co-occurring psychiatric diagnoses being anxiety disorders, Attention-Deficit/Hyperactivity Disorder (ADHD), disruptive, impulse-control, conduct disorders, and mood disorders [2,3,4]. While the gold standard of ASD symptom management includes behavioral, educational, and psychosocial interventions (i.e., Applied Behavioral Analysis-based therapies, parent management training, Individualized Education Programs in schools, social skills training, etc.), psychopharmacologic intervention has been included as one of the main components in the management of ASD associated problem behaviors and comorbidities [5]. Although there is no pharmacological treatment for the core symptoms of ASD, the US Food and Drug Administration (FDA) approved risperidone (2006) and later aripiprazole (2009) for irritability, including symptoms of aggressive behaviors. Despite this limited set of approved medications and indications, previous studies have shown increasing rates of psychotropic use and the simultaneous use of multiple psychotropic medications and polypharmacy among youth with ASD, especially if other treatments are not available or accessible [6].

Reported rates of psychotropic use among youth with ASD have varied widely. A recent systematic review indicates that rates of psychotropic use have ranged from 27% to 83% with psychotropic polypharmacy ranging from 35% to 54% [6,7]. This variability may be attributable to inconsistent operational definitions of psychotropic polypharmacy (PP) and multiclass polypharmacy (MPP) across studies [8].

Furthermore, many of the studies thus far have been based on parent reports or insurance-based claims data with relatively fewer studies focused on clinic-based prescribing practices. Studies done on prescribing patterns in clinic settings have shown large variations in prescribing psychotropic medication to children with ASD [9,10]. Previous studies have also revealed that ASD may be underdiagnosed in children of color and that there may also be differences in the rate of identification of problem behaviors in children of color and the use of psychotropic medications [11,12]. The existing literature has inconsistently described other demographic factors, in particular, ethnicity and/or race, insofar as their potential effect on prescribing patterns. When other demographic factors have been examined, there appears to be variability in the effects of insurance, age, sex, etc., on prescribing patterns [5,6,7,8,9,10,11,12].

Since the onset of the COVID-19 pandemic, there has been an observed rise in the rates of children reporting new-onset anxiety, depression, irritability, boredom, loneliness, inattention problems, and fear of COVID-19 [13,14], likely due to a combination of factors including pandemic-related lockdown procedures such as school closures, routine changes, caregiver burnout or the illness or death of family or caregivers, pandemic-related socioeconomic impacts, loss of access to mental health resources, etc. Emergency room visits related to child mental health concerns and suicide have also increased. In 2021, the American Academy of Pediatrics and the American Academy of Child and Adolescent Psychiatry declared a national emergency regarding childhood mental health [15,16]. Children with ASD may be particularly affected by pandemic-related lockdown procedures due to a low tolerance for uncertainty, disruption in routine, loss of access to therapy and other community resources, and loss of opportunities for developing important social and behavioral skills via peer interactions [17,18].

There is a relative dearth of literature examining psychotropic prescribing patterns in general for the COVID-19 pandemic. One study demonstrated increased fills for trazodone, decreased fills for benzodiazepines and hypnotics, and stable fills for antidepressants amongst adult patients during the early pandemic [19]. To our knowledge, there do not appear to be any studies thus far which have reviewed how prescription patterns have varied in the context of pandemic-related lockdowns and the general transition to outpatient telepsychiatry/telehealth services during the pandemic.

In this study, we have examined the prescribing patterns of an urban academic institute-affiliated tertiary autism center which provides assessment, diagnosis, treatment, and support for children and young adults with ASD. The primary aim of this study was to examine whether there is a meaningful change in psychotropic med prescribing patterns after the onset of the COVID-19 pandemic. We hypothesized that pandemic-related disruption in routines, loss of (or reduced) access to non-medical intervention, and reduced access to community resources and services, combined with worsening symptom severity and caregiver burnout would lead to an increase in polypharmacy rates and the use of select indication-based medication classes such as antipsychotics. We also hypothesized that there may be differences in prescribing patterns across select factors including age, sex, race/ethnicity, insurance, number of visits per calendar year, and comorbidities.

## 2. Materials and Methods

### 2.1. Study Participants

The present study provides a retrospective review of electronic health records (EHR) of patients seen at an urban academic institute-affiliated tertiary Autism Center Psychiatry Clinic. Inclusion criteria were (a) age between 3 and 21 years, (b) clinical diagnosis of ASD, (c) seen for psychiatric medication evaluation and/or management, (d) clinic visits occurred any period between 1 January 2019 and 31 December 2020, and (e) seen by either a psychiatrist or a psychiatric advanced registered nurse practitioner. Exclusion criteria were (a) younger than 3 years or older than 21 years, (b) without clinical diagnosis of ASD, and (c) visit reason not for medication evaluation and/or management. For convenience, the year 2019 was treated as a “pre-pandemic” year, and 2020 as a “pandemic” year. A total of 851 and 525 patients were seen in 2019 and 2020, respectively. Clinical characteristics of this “total clinic cohort” in each year are shown in Table 1. Among the patients seen in 2019, 473 patients returned to the clinic in 2020 (named as “paired-sample”) whereas the remaining 378 patients did not return to the clinic in 2020 (named as “not-paired sample”). To identify clinical factors associated with the likelihood of patients returning to the clinic in 2020, we compared the clinical characteristics of these paired and not-paired samples from the year 2019 (Table 2). Because the total clinic cohort is a naturalistic sample containing duplicate patients, we also created a separate dataset (“random unique sample”) containing only unique patients in each year by randomly assigning duplicate patients to one of the years (for example, if a patient was seen in both years, this patient was randomly assigned to either 2019 or 2020). The clinical characteristics of this random unique sample between 2019 and 2020 are shown in Table 3. This study was approved by the Institutional Review Board (IRB) with a waiver of Informed Consent and Health Insurance Portability and Accountability Act (HIPAA) authorization.

### 2.2. Study Procedures

Demographic, insurance, ICD-10 diagnostic codes associated with visits, and prescription data were extracted from the Electronic Health Records (EHR).

We examined the presence of the following commonly co-occurring psychiatric disorders using the International Classification of Diseases, Tenth Revision, Clinical Modification (ICD-10-CM) codes: anxiety disorders (F40, F41, F42, and F93.0), ADHD (F90), disruptive behavior spectrum (F34.81, F63.81, F91.3, F91.9, F98.9, R45.6, R45.850, R45.4, and R45.1), and depressive disorders (F32.1,F32.2, F32.3, F32.4, F32.5, F32.8, F32.9, F21.A, F33, and F34.1). We also presented the rates of intellectual disability (F70, F71, F72, F73, F78, and F79), but we suspect that the rates of intellectual disability may have been under-estimated as neuropsychologic testing (including cognitive, intellectual, and/or adaptive functioning) results were not always available.

Psychotropic medications in this study included medications prescribed for psychiatric, emotional, or behavioral health indications. Medications available over the counter (OTC) (e.g., melatonin, diphenhydramine, N-acetylcysteine, etc.) were excluded because most OTC medications are not prescribed. We also excluded medications prescribed to address the side effects of psychotropic medications, such as benztropine, trihexyphenidyl, cyproheptadine, or metformin from the subsequent analyses. Unique medications in this study included medications with the same active pharmaceutical ingredient, for example, divalproex sodium, valproate sodium, and valproic acid are treated as one unique medication. A total of 54 unique medications met the above definition of psychotropic medications in this study.

In this study, psychotropic polypharmacy (PP) was defined as ≥2 unique psychotropic medications prescribed in each calendar year (cumulative), without taking into consideration the overlap period of multiple medications. For example, a patient prescribed fluoxetine, risperidone, and olanzapine during the calendar year 2019 would have been considered as having been prescribed 3 unique psychotropic medications that year, regardless of whether these medications overlapped chronologically. Multiclass psychotropic polypharmacy (MPP) was defined as ≥2 psychotropic classes in each calendar year (cumulative), without taking into consideration the overlap period of different classes of medications. For MPP analysis, we grouped unique medications into 6 major categories: ADHD medications, antidepressants, antipsychotics, mood stabilizers, anxiolytics, and hypnotics (See Appendix A), and medications not included in the above categories were not analyzed for MPP. Antiepileptic medications, such as valproic acid and lamotrigine, were classified as mood stabilizers as they were prescribed by psychiatrists or psychiatric advanced registered nurse practitioners, though some of them may have served dual purposes. Using the previous example, a patient prescribed fluoxetine, risperidone, and olanzapine during the calendar year 2019 would have been considered as having been prescribed 2 unique psychotropic medication classes that year, regardless of whether these medications overlapped chronologically.

### 2.3. Statistical Analyses

The IBM SPSS^®^ Statistical Software (version 27, IBM Corp., Armonk, NY, USA) was used for analyses. Descriptive statistics were used to describe the characteristics of the clinic sample, which included age (grouped to preschoolers (3 to 5 years), preteens (6 to 12 years), teens (13 to 17 years), and young adults (18 to 21 years)), sex assigned at birth, self-identified ethnicity (Hispanic and non-Hispanic), race (Asian, Black, White, 2 more races, and Other (Other includes Native American, Pacific Islander, and self-identified as “other” due to small numbers in each category)), types of insurance (Commercial, Medicaid, and Other), four major psychiatric diagnoses (Anxiety disorders, ADHD, disruptive behavior disorders, and depressive disorders), and six classes of psychotropic medications prescribed (ADHD medications, antidepressants, antipsychotics, mood stabilizers, anxiolytics, and hypnotics), number of visits, number of unique medications, number of medication classes, presence/absence of PP and MPP, per each index year during the study period. Chi-squared statistics were used to examine the difference in clinical characteristics (categorical variables) between sample groups (e.g., total clinic cohort for 2019 vs. 2020, paired vs. not-paired sample, and unique patients for 2019 vs. 2020). In Table 1, the total clinic cohorts seen in 2019 and 2020 were treated as if they were independent (even though they include duplicate individuals) for the purpose of comparison only. A random unique sample set was created using the SPSS random case selection function. A non-parametric, Mann–Whitney test was used to examine whether continuous variables (number of four major diagnoses, number of visits, unique medications, and medication classes) differ between “paired” and “not-paired” samples in 2019 (Table 4). Information as “Unknown” was treated as missing and excluded from the subsequent statistical analyses.

Binary logistic regression analyses were also performed to ascertain the effects of paired-ness (categorical), age (continuous), sex (categorical), ethnicity (categorical), race (categorical), insurance (categorical), and number of visits during the index year (continuous) on the likelihood of presence/absence of following dependent variables: PP and MPP in each index year (2019 and 2020) (Table 5).

Study-wide significance was set as nominal *p*-value less than 0.05 without correcting for multiple comparisons.

## 3. Results

A total of 851 patients were seen for psychiatric medication management in 2019. In 2020, 525 patients were seen, with 90% (473 out of 525) of this 2020 sample representing patients who were also seen in 2019. There were no significant differences in the total clinic cohort between 2019 and 2020 for age group, sex, ethnicity, race, or type of insurance; however, the total clinic cohort seen in 2020 had a higher portion of anxiety (77% vs. 65.5%, *p* < 0.001), ADHD (70.5% vs. 59.2%, *p* < 0.001), and disruptive behavior diagnoses (60.8% vs. 50.5%, *p* < 0.001), higher rates of antidepressant prescription than patients seen in 2019 (62.9% vs. 56.9%, *p* = 0.028), and higher rates of MPP (67.8% vs. 61.9%, *p* = 0.027), when we treated each year cohort as “independent” for the purpose of comparison in this study (Table 1). The number of visits, unique medications prescribed, and number of medication classes were described for the total clinic cohort in 2019 vs. 2020 (Appendix A), and for not-paired vs. paired-samples in 2019 (Appendix A). The rates of PP in the total clinic cohort were 71.6% and 75.6% in 2019 and 2020, respectively. The rate of MPP in the total clinic cohort were 61.9% and 67.8% in 2019 and 2020, respectively.

Compared to the not-paired sample (*n* = 378), the paired-sample (*n* = 473) were more likely to be in the age group of 13–17 years (39.7% vs. 31.0%, *p* < 0.001), self-identified as non-Hispanic (85.4% vs. 77.8%, *p* = 0.016), diagnosed with anxiety (78.9% vs. 48.7%, *p* < 0.001), ADHD (71.0% vs. 44.4%, *p* < 0.001), depression (23.9% vs. 13.0%, *p* < 0.001) and disruptive behavior (63.3% vs. 33.3%, *p* < 0.001) diagnoses, higher rates of antidepressants (63.4% vs. 48.7%, *p* < 0.001), ADHD medications (72.5% vs. 59.8%, *p* < 0.001), and antipsychotics (36.8% vs. 26.2%, *p* < 0.001) prescriptions, and higher rates of PP (81.6% vs. 59.0%, *p* < 0.001) and MPP (71.0% vs. 50.5%, *p* < 0.001) than the not-paired sample (Table 2). These clinical factors are associated with the likelihood of patients returning to the clinic in the pandemic year.

Similarly, the random unique sample showed higher rates of anxiety (75.0% vs. 60.2%, *p* < 0.001), ADHD (69.9% vs. 54.6%, *p* < 0.001), and disruptive behavior (57.9% vs. 45.4%, *p* < 0.001) diagnoses in 2020 vs. 2019, but PP or MPP rates did not differ across two years (Table 3).

The paired-sample also had a higher number of comorbid psychiatric diagnoses, clinic visits, unique medications, and medication classes prescribed in 2019 (Table 4). In addition, 32.1% and 12.1% of the paired-sample were on 4 or more medications and 4 or more medication classes, whereas 18.8% and 9.0% of the not-paired sample were on 4 or more medications and 4 or more medication classes (Appendix A). This “paired” effect and number of visits were significant for both years for PP (*p* < 0.001 in 2019 and 2020) and MPP (*p* < 0.001 in 2019, *p* = 0.006 in 2020) in the binary logistic regression analysis (Table 5). When we examined the distribution of select demographic factors for PP and MPP, age was significantly associated with increased rates of PP (*p* = 0.002) and MPP (*p* < 0.001) in 2019 and MPP in 2020 (*p* = 0.046), but not for PP in 2020 (*p* = 0.067) (Appendix A). Race was only significant in 2019 for both PP (*p* = 0.042) and MPP (*p* = 0.03) with Asians less likely to be on polypharmacy. Notably, a substantial proportion of patients received 4 or more unique psychotropic medications in each year with over 26% receiving 4 or more medications and approximately 10% receiving 4 or more medication classes in each year (Appendix A).

The most prescribed medication class in both 2019 and 2020 was ADHD medication (33.5% in both years), followed by antidepressants (28.5% and 29.9%, respectively) and antipsychotics (16.1% and 15.2%, respectively) (Appendix A). In 2019, the most prescribed medication was methylphenidate and guanfacine in 2019 and 2020, respectively (Appendix A). Across both years, the top 5 prescribed medications were methylphenidate-based formulations, guanfacine, sertraline, clonidine, and fluoxetine. Risperidone was the 6th most prescribed medication in each year, whereas the 7th most prescribed medication was aripiprazole in 2019 and hydroxyzine in 2020.

## 4. Discussion

In this study, we observed high rates of PP and MPP with over 70% of patients receiving more than one psychotropic medication and over 60% of patients on more than one psychotropic medication class. These polypharmacy rates were higher than those previously reported [6,7], likely due to (a) the broad definition of PP and MPP in this study that did not take into consideration the overlap period in each study year and (b) the study sample drawn from a tertiary autism center that serves patients with higher clinical acuity and severity. Given the wide variability in reported rates of polypharmacy, it would be important to use a consistent definition across studies.

We observed no significant differences of PP and MPP rates in race, ethnicity, sex, or insurance status. Age was positively associated with an increased rate of PP and MPP (Appendix A), consistent with other studies [6,7]. While youth who self-identified as Black did seem to exhibit a higher proportion of PP and MPP compared to other racial groups identified in 2019 (80.9% exhibited PP and 66.7% exhibited MPP in 2019), the differences in PP and MPP between racial groups was not statistically significant (Appendix A). Moreover, given over 60% of our sample self-identified as White, with a relatively limited proportion identifying as non-White, an analysis of the influence of race is limited in this study. Likewise, our sample skewed towards majority male, commercially insured, and non-Hispanic. We would continue to encourage future analyses of the impact of demographic factors on prescribing patterns in clinical settings.

Overall, the impact of the 2019 COVID-19 pandemic was mixed in this study. While we did observe a small increase in PP (71.6% to 75.6%) and MPP rates (61.9% to 67.8%) from 2019 to 2020 in our total clinic cohort, only the latter was nominally significant (*p* = 0.027). Notably, statistically significant increases in rates of anxiety, ADHD, and disruptive behavior diagnoses were noted in all three analyses including the total clinic cohort (2020 vs. 2019), the paired-sample vs. not-paired sample, and the random unique sample (2020 vs. 2019), suggesting that psychiatric comorbidity is associated with the likelihood of returning to the clinic during the pandemic. Notably, being seen again in 2020 (paired-sample) was a predictor for the likelihood of PP or MPP (Table 5). This difference between our paired and not-paired sample may suggest increased clinical needs, acuity, and/or severity in the patients seen in both years (as opposed to just in 2019 with no follow-up in 2020). Another possible explanation involves changes related to the COVID-19 pandemic, such as reduced access to mental health resources, educational services, and general disruption in routines, though it remains difficult to infer this directly from our available data. While these differences in comorbidity rates and prescribing patterns may be associated with the pandemic, our study was not able to identify the direct impact of the pandemic on prescribing patterns and polypharmacy rates given the nature of a retrospective chart review.

Although, we hypothesized that patients seen in 2020 may have higher rates of antipsychotic utilization due to decreased access to healthcare resources, school and other therapeutic services, disruptions in routine, and increased household and financial stress owing to the COVID-19 pandemic, we observed no significant differences in the proportion of antipsychotic medications prescribed in the total clinic cohort. Nevertheless, approximately 32% of patients are being prescribed at least one antipsychotic in each year (Table 1). This rate is higher than other studies which found that approximately 17% of children with ASD are prescribed at least one antipsychotic [12,20]. The overall high rate of antipsychotic prescription may be related to the study location at a tertiary autism center.

In our random unique sample, we observed higher rates of anxiety (75.0% vs. 60.2%, *p* < 0.001), ADHD (69.9% vs. 54.6%, *p* < 0.001), and disruptive behavior (57.9% vs. 45.4%, *p* < 0.001) in 2020 although PP or MPP rates did not differ from 2019 to 2020 (Table 3). Given that our study period ended in December 2020, it is possible that the effects on prescribing patterns were yet to emerge. While we may at least infer that this supports the hypothesis that the pandemic likely increased rates of mental health and behavioral problems in youth with ASD, studies utilizing independent samples extending into the years beyond 2020 are needed to come to this conclusion.

A surprisingly high number of patients were prescribed ≥4 medications per calendar year in this study. Twenty-six percent of patients in 2019 and 2020 were prescribed ≥4 medications and approximately 10% in each year were prescribed ≥4 medication classes. While we continue to observe high rates of psychotropic use, PP and MPP, the potential benefit of the use of multiple psychotropic agents in either the treatment of core symptoms or comorbidities in children with ASD remain unclear. In certain situations, such as in cross titration, the treatment of multiple co-morbid conditions, or the short-term addition of antipsychotic medications, polypharmacy may be determined to be of clinical benefit or utility by clinicians. However, concerns may continue to exist among patients, parents, and clinicians regarding the potential impact of such medications on developing brains, particularly in individuals with ASD, possible medication interactions, and the susceptibility of individuals with ASD to medication side effects.

Of note, the most prescribed class of medication in this study was ADHD medication (33.5% in both years), followed by antidepressants (28.5% and 29.9%, respectively) and antipsychotics (16.1% and 15.2%, respectively) (Appendix A). Antipsychotics, while still among the most prescribed medications, were utilized relatively less as a class than seen in some other studies [7]. While this could be an indication of changing prescribing patterns overall given concerns about long-term antipsychotic use, given our sample was limited to a single tertiary center, this may simply reflect the prescribing preferences of the clinicians in the setting examined.

Our study has several limitations given our naturalistic sample and retrospective analysis. First, our definition of polypharmacy was based on a cumulative number of prescribed medications in each calendar year without accounting for the possible overlap. For example, it is possible that a patient tried a medication for a few weeks, and then tried a second medication due to an adverse reaction to the first medication. Given the nature of our study, we were not able to clarify this question. Second, the total clinic cohort in 2019 and 2020 was not entirely independent as the majority of 2020 patients were seen in both years, although we treated the two sample groups independently for the purpose of comparison. Therefore, the results presented in Table 1 should be interpreted with caution. To compensate, we conducted additional data analyses for paired vs. not-paired samples as well as for the random unique sample set. Third, while we attempted to find the effects of the pandemic on prescribing patterns by comparing 2019 and 2020, we did not exclude data that could be considered “pre-pandemic” from the 2020 sample because pandemic-related anxiety and changes happened gradually and in different timelines across the county, within mental health agencies, and within school districts, making it difficult to pinpoint the onset. While only 40 patients would have been excluded from the 2020 sample if we had shortened our pandemic sample to the start of March 2020, it remains possible that the inclusion of this sample in the pandemic group influenced our results. Fourth, while we linked each prescribed medication record to individual psychiatry clinic visits, we could not completely rule out medications prescribed for non-psychiatry or non-behavioral purposes (i.e., mood stabilizers used for seizure-based indications). Fifth, while we included rates of intellectual disability in this study based on associated ICD-10 codes, due to the limited availability of neuropsychologic testing data, these estimates may be imprecise. Sixth, in this study, we were unable to account for family access to other behavioral, healthcare, educational, or other psychosocial resources. Further study is also necessary to investigate whether the number of psychotropic medications are inversely correlated with access to other services in the community (e.g., in-home applied behavior analysis, patient and family counseling/psychotherapy, etc.), the severity of core symptoms, and the presence of problem behaviors. Lastly, this study used indication-based psychotropic classes to reflect clinical practice for the purpose of this study. However, we recognize that neuroscience-based nomenclature (NbN) has become more commonplace and may be the basis of future MPP operational definitions [21].

## 5. Conclusions

This study found high rates of PP and MPP in our study sample. Further study is needed regarding the effectiveness and safety of the concurrent use of multiple unique medications or medication classes in youth with ASD in both the treatment of core symptoms and related common comorbidities. Such research may guide a more standardized operational definition of PP and MPP and help to guide safe prescribing habits in clinical settings. Likewise, further study should be done regarding the potential influences of demographic factors such as race and ethnicity, socioeconomic status, and other factors such as access to behavioral, educational, or other psychosocial resources on prescribing patterns and polypharmacy rates in clinical settings. Such research is important in examining potential biases or influences on the medical prescribing practices to youth with ASD and ensuring equitable care and treatment. While higher rates of anxiety, ADHD, and disruptive behavior diagnoses may be associated with the pandemic, we suggest further studies utilizing independent samples to confirm the factors impacting prescribing patterns and polypharmacy rates in youth with ASD during the COVID-19 pandemic. Overall, there were minimal observed changes in prescribing habits from 2019 to 2020. However, further study is needed regarding the potential impacts of the COVID-19 pandemic (including in years extending beyond 2020) and related restrictions or limitations in access to resources, therapies, and treatments.

## Figures and Tables

**Table 1 jcm-11-04855-t001:** Clinical Characteristics of the “total clinic cohort” in 2019 and 2020.

		2019 (*n* = 851)	2020 (*n* = 525) ^1^	Statistics ^2^
		*n* (%)	*n* (%)	Chi (df), *p*-Value
Age group	3 to 5 years	18 (2.1)	6 (1.1)	5.413 (3), ns
	6 to 12 years	371 (43.6)	204 (38.9)	
	13 to 17 years	305 (35.8)	206 (39.2)	
	18 to 21 years	157 (18.4)	109 (20.8)	
Sex	Male	683 (80.3)	410 (78.1)	0.930 (1), ns
	Female	168 (19.7)	115 (21.9)	
Ethnicity	Hispanic	95 (12.0)	51 (10.2)	0.922 (1), ns
	Non-Hispanic	698 (88.0)	447 (89.8)	
Race	White	536 (66.3)	340 (66.7)	0.987 (4), ns
	Asian	69 (8.5)	49 (9.6)	
	Black	42 (5.2)	28 (5.5)	
	2 or more	64 (7.9)	37 (7.3)	
	All other ^3^	98 (12.1)	4 (11.0)	
Insurance	Commercial	545 (64.0)	339 (64.6)	0.716 (2), ns
	Medicaid	280 (32.9)	174 (33.1)	
	Other	26 (3.1)	12 (2.3)	
Anxiety spectrum		557 (65.5)	404 (77.0)	20.386 (1), <0.001
ADHD ^4^		504 (59.2)	370 (70.5)	17.739 (1), <0.001
Disruptive behavior spectrum		430 (50.5)	319 (60.8)	13.708 (1), <0.001
Depression spectrum		162 (19.0)	117 (22.3)	2.121 (1), ns
Intellectual Disability		124 (14.6)	89 (17.0)	1.407 (1), ns
ADHD medications		569 (66.9)	370 (70.5)	1.956 (1), ns
Antidepressants		484 (56.9)	330 (62.9)	4.810 (1), 0.028
Antipsychotics		273 (32.1)	168 (32.0)	0.001 (1), ns
Mood stabilizers		124 (14.6)	78 (14.9)	0.021 (1), ns
Anxiolytics		184 (21.6)	117 (22.3)	0.084 (1), ns
Hypnotics		65 (7.6)	40 (7.6)	0.000 (1), ns
PP ^5^		609 (71.6)	397 (75.6)	2.717 (1), ns
MPP ^6^		527 (61.9)	356 (67.8)	4.887 (1), 0.027

^1^ includes the same individual patients (*n* = 473) seen in the year 2019; ^2^ Treated samples from 2019 and 2020 as independent for comparison purpose only; ^3^ includes American Natives, Pacific Islanders, and self-identified as “Other”; ^4^ Attention-Deficit/Hyperactivity Disorder; ^5^ Psychotropic polypharmacy defined as ≥2 unique psychiatric medications; ^6^ Multiclass Psychotropic Polypharmacy defined as ≥2 psychotropic classes; ns—not significant (nominal significance was set as *p*-value < 0.05).

**Table 2 jcm-11-04855-t002:** Comparison of Clinical characteristics between paired * vs. not-paired ** samples in 2019.

		Not-Paired (*n* = 378)	Paired (*n* = 473)	Statistics
		*n* (%)	*n* (%)	Chi (df), *p*-Value
Age group	3 to 5 years	11 (2.9)	7 (1.5)	16.295 (3), <0.001
	6 to 12 years	161 (42.6)	210 (44.4)	
	13 to 17 years	117 (31.0)	188 (39.7)	
	18 to 21 years	189 (23.5)	68 (14.4)	
Sex	Male	308 (81.5)	375 (79.4)	0.642 (1), ns
	Female	70 (18.5)	98 (20.7)	
Ethnicity	Hispanic	52 (13.8)	43 (9.1)	8.307 (2), 0.016
	Non-Hispanic	294 (77.8)	404 (85.4)	
Race	White	233 (66.6)	303 (66.0)	3.953 (4), ns
	Asian	24 (6.9)	45 (9.8)	
	Black	16 (4.6)	26 (5.7)	
	2 or more	32 (9.1)	32 (7.0)	
	All other ^1^	45 (12.9)	53 (11.5)	
Insurance	Commercial	230 (60.8)	315 (66.6)	4.145 (2), ns
	Medicaid	138 (36.5)	142 (30.0)	
	Other	10 (2.6)	16 (3.4)	
Anxiety spectrum		184 (48.7)	373 (78.9)	84.635 (1), <0.001
ADHD ^2^		168 (44.4)	336 (71.0)	61.519 (1), <0.001
Disruptive behavior spectrum		126 (33.3)	304 (64.3)	80.444 (1), <0.001
Depression spectrum		49 (13.0)	113 (23.9)	16.276 (1), <0.001
Intellectual Disability		38 (10.1)	86 (18.2)	11.153 (1), 0.001
ADHD medications		226 (59.8)	343 (72.5)	15.361 (1), <0.001
Antidepressants		184 (48.7)	300 (63.4)	18.630 (1), <0.001
Antipsychotics		99 (26.2)	174 (36.8)	10.826 (1), <0.001
Mood stabilizers		50 (13.2)	74 (15.6)	0.986 (1), ns
Anxiolytics		69 (18.3)	115 (24.3)	4.551 (1), 0.033
Hypnotics		31 (8.2)	37 (7.2)	0.306 (1), ns
PP ^3^		223 (59.0)	386 (81.6)	52.787 (1), <0.001
MPP ^4^		191 (50.5)	336 (71.0)	37.474 (1), <0.001

* Patients seen in both 2019 and 2020; ** patients seen in 2019 but not in 2020; ^1^ includes American Natives, Pacific Islanders, and self-identified as “Other”; ^2^ Attention-Deficit/Hyperactivity Disorder; ^3^ Psychotropic polypharmacy defined as ≥2 unique psychiatric medications per each calendar year period; ^4^ Multiclass Psychotropic Polypharmacy defined as ≥2 psychotropic classes per each calendar year period; ns—not significant (nominal significance was set as *p*-value < 0.05).

**Table 3 jcm-11-04855-t003:** Clinical characteristics of the “random unique sample” *.

		2019 (*n* = 606)	2020 (*n* = 292)	Statistics
		*n* (%)	*n* (%)	Chi (df), *p*-Value
Age group	3 to 5 years	12 (2.0)	5 (1.7)	0.488 (3), ns
	6 to 12 years	259 (42.7)	119 (40.8)	
	13 to 17 years	216 (35.6)	110 (37.7)	
	18 to 21 years	119 (19.6)	58 (19.9)	
Sex	Male	504 (83.2)	213 (72.9)	12.797 (1), <0.001
	Female	102 (16.8)	79 (27.1)	
Ethnicity	Hispanic	73 (13.0)	29 (10.5)	2.560 (2), ns
	Non-Hispanic	489 (87.0)	248 (89.5)	
Race	White	372 (61.4)	197 (67.5)	12.037 (4), 0.017
	Asian	42 (6.9)	31 (10.6)	
	Black	35 (5.8)	10 (3.4)	
	2 or more	47 (7.8)	20 (6.8)	
	All other ^1^	110 (18.2)	34 (11.6)	
Insurance	Commercial	374 (61.7)	198 (67.8)	3.730 (2), ns
	Medicaid	214 (35.3)	89 (30.5)	
	Other	18 (3.0)	5 (1.7)	
Anxiety spectrum		365 (60.2)	219 (75.0)	18.901 (1), <0.001
ADHD ^2^		331 (54.6)	204 (69.9)	19.0101), <0.001
Disruptive behavior spectrum		275 (45.4)	**169 (57.9)**	**12.312 (1), <0.001**
Depression spectrum		101 (16.7)	63 (21.6)	3.181 (1), 0.075
Intellectual Disability		86 (14.2)	41 (14.0)	0.004 (1), ns
ADHD medications		398 (65.7)	205 (70.2)	1.832 (1), ns
Antidepressants		332 (54.8)	175 (59.9)	2.123 (1), ns
Antipsychotics		188 (31.0)	87 (29.8)	0.140 (1), ns
Mood stabilizers		87 (14.4)	39 (13.4)	0.163 (1), ns
Anxiolytics		113 (18.6)	64 (21.9)	1.332 (1), ns
Hypnotics		48 (7.9)	21 (7.2)	0.148 (1), ns
PP ^3^		412 (68.0)	211 (72.3)	1.694 (1), ns
MPP ^4^		359 (59.2)	186 (63.7)	1.641 (1), ns

* No duplicate subjects are included in this sample set; ^1^ includes American Natives, Pacific Islanders, and self-identified as “Other”; ^2^ Attention-Deficit/Hyperactivity Disorder; ^3^ Psychotropic polypharmacy defined as ≥2 unique psychiatric medications per each calendar year period; ^4^ Multiclass Psychotropic Polypharmacy defined as ≥2 psychotropic classes per each calendar year period; ns—not significant (nominal significance was set as *p*-value < 0.05).

**Table 4 jcm-11-04855-t004:** Mann–Whitney Tests for paired * vs. not-paired ** samples in 2019.

	Not-Paired (*n* = 378)	Paired (*n* = 473)	
	Mean (SD)	Mean (SD)	*p*-Value
No. comorbidity ^1^	1.39 (1.14)	2.38 (0.88)	<0.001
No. Visits	2.35 (1.659)	3.50 (1.914)	<0.001
No. of unique meds	2.16 (1.543)	2.92 (1.613)	<0.001
No. of classes	1.74 (1.154)	2.20 (1.065)	<0.001

* Patients seen in both 2019 and 2020; ** patients seen in 2019 but not in 2020; ^1^ indicates number of four major psychiatric comorbidities including ADHD, anxiety, depression, and disruptive behaviors.

**Table 5 jcm-11-04855-t005:** Binary Logistic Regression Model of Psychotropic Prescriptions and Patient Characteristics.

	2019		2020	
	PP	MPP	PP	MPP
	Chi (df), *p*-Value	Chi (df), *p*-Value	Chi (df), *p*-Value	Chi (df), *p*-Value
Overall Model	139.693 (11), <0.001	119.065 (11), <0.001	114.268 (11), <0.001	88.567 (11), <0.001
Paired	15.165 (1), <0.001	12.897 (1), <0.001	19.646 (1), <0.001	8.996 (1), =0.003
Age	13.690 (1), <0.001	33.006 (1), <0.001	ns	6.210 (1), =0.013
Sex	ns	ns	ns	ns
Ethnicity	ns	ns	ns	ns
Race	9.297 (4), ns (0.054) Asian *p* = 0.042 (−) Black *p* = ns (0.070) (+)	6.921 (4), ns Asian *p* = 0.030 (−) Black *p* = ns	ns	ns
Insurance	ns	ns	ns	ns
Visit No/year	52.688 (1), <0.001	36.879 (1), <0.001	53.651 (1), <0.001	47.556 (1), =0.006

Df: degree of freedom; (−) Negatively associated; (+) Positively associated; ns—not significant (nominal significance was set as *p*-value < 0.05). PP—psychotropic polypharmacy, MPP—multiclass psychotropic polypharmacy.

## Data Availability

The data are not publicly available due to patient privacy.

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
