# Peer review of "A Retrospective Chart Review of Factors Impacting Psychotropic Prescribing Patterns and Polypharmacy Rates in Youth with Autism Spectrum Disorder during the COVID-19 Pandemic"

_jcm, 2022, doi:10.3390/jcm11164855_

Round 1

Reviewer 1 Report

Dear authors…

In general the paper is well written but needs spelling and grammar check.

Comments:

·         The abstract should be rewritten because it doesn’t give the reader any idea about the work you did. Amis should be clearly mentioned or at least the main aim should.

·        Significant results should be mentioned in the abstract and result section, the word significant or insignificant must be followed by p-value even though it is mentioned in the tables. Not only percentages.

·         In the analysis, the patients were analyzed as independent although most of the patients in 2019 were seen in 2020. I believe that the author should try to do the analysis for that patients as dependent this will give a very good insight about the effect of the medications and their safety in the same patient.

Author Response

Response to Reviewer 1 Comments

In general the paper is well written but needs spelling and grammar check. à Thank you for this comment. To our knowledge, all spelling and grammar errors have been corrected.  

Comments:

  • The abstract should be rewritten because it doesn’t give the reader any idea about the work you did. Amis should be clearly mentioned or at least the main aim should.
  • Response: We revised the abstract and included the aims of the study as “This study is to examine the factors associated with psychotropic prescribing patterns, including rates of PP and multiclass polypharmacy (MPP) in youth with ASD during the COVID-19 pandemic.

  • Significant results should be mentioned in the abstract and result section, the word significant or insignificant must be followed by p-value even though it is mentioned in the tables. Not only percentages.
  • Response: p-values are now included in the abstract and result section.

  • In the analysis, the patients were analyzed as independent although most of the patients in 2019 were seen in 2020. I believe that the author should try to do the analysis for that patients as dependent this will give a very good insight about the effect of the medications and their safety in the same patient.
  • Response: Thank you! We pondered a lot about how to examine the clinical characteristics across 2019 and 2020 as this naturalistic sample partially (not entirely) contained duplicate individuals. Because of this, we further analyzed our sample by comparing the clinical characteristics of patients who returned to the clinic in 2020 vs patients who did not return to the clinic in 2020 (Table 2), as we anticipated that those seen again in 2020 may have more psychiatric/behavioral challenges. In addition, we also added another analysis by randomly assigning duplicate individuals to either year 2019 or 2020, creating a random unique sample set (Table 3). Therefore, individual patient will only appear once. These three tables (1 to 3) showed partially concordant results. These were included in the revised manuscript.

Reviewer 2 Report

The authors present an analysis of psychotropic and non-psychotropic (retrospective) prescribing patterns and diagnoses in people aged between 3 to 21 years with autism using electric records from a USA tertiary academic autism treatment center.  A comparison is made between the period that approximates to the pre-pandemic (2019) and pandemic epoch (2020).  One of the findings was that, “Patients seen in 2020 had higher rates of anxiety, attention deficit hyperactivity disorder, and disruptive behavior diagnoses; and higher rates of antidepressant prescription than patients seen in 2019.”

The introduction and study limitations are okay, and the study methods are satisfactory outlined. The statistical analysis is used correctly.  

I suggest:

(1) to clarify the definitions of PP and MPP by giving examples.

(2) to include data regarding the number of patients with an intellectual disability (if available).

(3) the title be changed to put a greater emphasis on the key finding: the impact of the pandemic on clinical outcomes in this group.

(4) in the introduction to emphasize that the mainstay of treatment of autism  (inc. core symptoms) is non-pharmacological.

(5) in the discussion more commentary or thoughts on the clinical impact of polypharmacy - are we prescribing too readily given the lack of published evidence for treatments and are we placing too little emphasis on the psychosocial domain. Is there an expectation from patients and their family of a preference for a medication solution?

Author Response

Dear Reviewer 2: 

(1) to clarify the definitions of PP and MPP by giving examples.

  • Thank you. This was included in our revision: For PP, “For example, a patient prescribed fluoxetine, risperidone and olanzapine during calendar year 2019 would have been considered as having been prescribed 3 unique psychotropic medications that year, regardless of whether these medications overlapped chronologically.” For MPP “Using the previous example, a patient prescribed fluoxetine, risperidone, and olanzapine during calendar year 2019 would have been considered as having been prescribed 2 unique psychotropic medication classes that year, regardless of whether these medications overlapped chronologically.”

(2) to include data regarding the number of patients with an intellectual disability (if available).

  • Thank you. We do have the frequency data for ID based on ICD-10 codes. We added a row for ID in Tables 1 to 3, and added following sentence, “We also presented the rates of intellectual disability (F70,F71,F72,F73,F78, and F79) in our sample, but we suspect that the rates of intellectual disability may have been under-estimated as neuropsychologic testing (including cognitive, intellectual and/or adaptive functioning) results were not always available.” Because of this, we did not further comment in our results section other than in Tables though we added this to our discussion “Fifth, while we included rates of intellectual disability in this study based on ICD10 codes, as previously noted, as we did not have neuropsychologic testing data readily available, these estimates may be imprecise.”

(3) the title be changed to put a greater emphasis on the key finding: the impact of the pandemic on clinical outcomes in this group.

  • Thank you for this suggestion. Title was changed to “A Retrospective Chart Review of Factors Impacting Psychotropic Prescribing Patterns and Polypharmacy Rates in Youth with Autism Spectrum Disorder During the COVID-19 Pandemic”

(4) in the introduction to emphasize that the mainstay of treatment of autism  (inc. core symptoms) is non-pharmacological.

  • We added following sentence in page 2, ” While the gold standard of ASD symptom management includes behavioral, educational and psychosocial interventions (i.e. Applied Behavioral Analysis based therapies, parent management training, Individualized Education Programs in schools, social skills training, etc), psychopharmacologic intervention has been included as one of main components in the management of ASD associated problem behaviors and comorbidities”

(5) in the discussion more commentary or thoughts on the clinical impact of polypharmacy - are we prescribing too readily given the lack of published evidence for treatments and are we placing too little emphasis on the psychosocial domain. Is there an expectation from patients and their family of a preference for a medication solution?

  • Thank you for this thoughtful comments! We agree completely with you on this. We added following paragraph in our discussion, “While we continue to observe high rates of psychotropic polypharmacy, the potential benefit of the use of multiple psychotropic agents in either the treatment of core symptoms or comorbidities in children with ASD remains unclear. In certain situations, such as in cross titration, the treatment of multiple co-morbid conditions, or the short-term addition of antipsychotic medications, polypharmacy may be determined to be of clinical benefit or utility by clinicians. However, concerns may continue to exist among patients, parents and clinicians regarding the potential impact of such medications on developing brains particularly in individuals with ASD, possible medication interactions, and the susceptibility of individuals with ASD to medication side effects.”

We hope this study would provide some valuable information and perspective to our fellow clinicians!

Round 2

Reviewer 1 Report

I would like to thank the authors for their work they added to the paper. It looks much better. Wish them all the best. 

Author Response

Thank you to Reviewer 1! We feel your comments helped us tremendously in improving our paper!  

Reviewer 2 Report

I have no further comments.

Author Response

Thank you to Reviewer 2!  We feel your comments helped us tremendously in improving our paper, we appreciate your time.  
